# TRAINING VARIABLE LONG SEQUENCES WITH DATA-CENTRIC PARALLEL

## ABSTRACT

Training deep learning models on variable long sequences poses significant computational challenges. Existing methods force a difficult trade-off between efficiency and ease-of-use. Simple approaches use static configurations that cause workload imbalance low efficiency, while complex methods introduces significant complexity and code change for new models. To break this trade-off, we introduce Data-Centric Parallel (DCP). Its core principle is to let the data itself drive the runtime. It achieves this by dynamically adjusting direct runtime settings (e.g., parallel size, gradient accumulation, recomputation) based on each batch's sequence length. Empirical results demonstrate that our method achieves up to a 2.88× speedup on 32 H200 GPUs. Designed for generalization, it can be integrated into any model with 10 lines of code. We anticipate this simple yet effective approach will serve as a robust baseline and facilitate future advancements in distributed training for variable long sequences.

## 1 INTRODUCTION

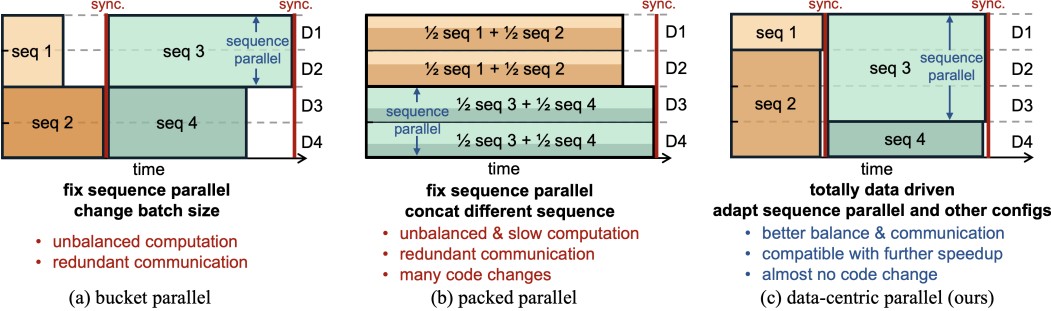

Figure 1: Comparison of parallel methods for variable sequences training including bucket parallel, packed parallel and data-centric parallel (ours). $Di$ refers to the $i$-$th$ device.

The capacity to process long sequences is a crucial driver for a growing number of deep learning applications. This trend is evident across diverse fields, including video generation (Brooks et al., 2024; Zheng et al., 2024; Polyak et al., 2024; Kong et al., 2024), image generation (Esser et al., 2024), multi-modal perception (Chen et al., 2024b; Wang et al., 2024), text generation (Touvron et al., 2023; Bai et al., 2023), and scientific computing (Jumper et al., 2021).

However, training on such data presents significant computational challenges: 1) **Long sequence**: the substantial length of sequences consumes lots of GPU memory, requiring sequence parallelism to partition one sequence across multiple devices to reduce the memory cost. 2) **Variable length**: The inherent wide variation in sequence lengths, as illustrated in Figure 3, leads to severe workload imbalances during distributed training, especially when combined with sequence parallel.

Parallel methods for training on variable-length sequences can be categorized into three classes as shown in Figure 1. Bucket parallel (Esser et al., 2024) sets a fixed parallel size for all sequences, and decrease the batch size for slow batches to reduce the imbalance, as indicated in Figure 2. Nevertheless, this naive solution introduces two problems. First, the batch size for very long sequences is often inherently limited, providing minimal flexibility for tuning and achieving the desired load

balance. Second, it does not consider the computational efficiency of short sequences when reducing batch size, leaving significant speed loss.

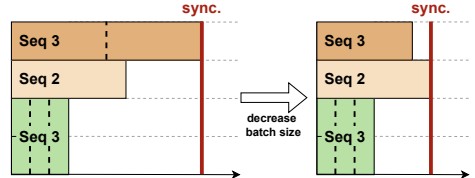

Packed parallel (Dehghani et al., 2023) improves load balance by packing multiple sequences into a batch instead of reducing batches, but this introduces communication overhead and requires extra changes for sequence level operations. Compiler-based parallel methods (Ge et al., 2024) apply a compiler to

Figure 2: Workload balance in bucket parallel for various sequence lengths. Dashed lines indicate the batch size.

automatically optimize for efficient plans, but this requires significant code changes and makes new model adaptation difficult.

Existing methods present a difficult trade-off between efficiency and ease-of-use and. We argue that the core limitation that preventing simple works from being effective is their dependence on predetermined runtime settings (e.g., parallel size), which significantly limits both communication cost and workload imbalance. By directly confronting this, it is possible to attain high efficiency without the need of more heavyweight systems. This leads us to the central question of our work: **how can we let the data itself drive the runtime in a simple yet effective way?**

To address this challenge, we propose Data-Centric Parallel (DCP), the first method that achieve both efficient and ease-of-use method for training variable long sequences. Instead of using a fixed setting, DCP dynamically adjusts the runtime settings such as parallel strategy, gradient accumulation, and recomputation based each data's sequence length. By minimizing the communication cost for each batch while balancing the workload, DCP significantly improves training throughput.

DCP comprises of two strategies: DCP-inter and DCP-intra. DCP-inter utilizes gradient accumulation to balance workload instead of reducing batch size. DCP-intra further speedup by minimizing recomputation. The extra memory cost is tackled by carefully adjusting the sequence parallel and batch size with ignorable cost.

Empirical results show that DCP effectively improve throughput for variable long sequences training by up to 2.88× with 32 H200 GPUs across 3 datasets on 2 models. Designed for generalization, DCP can be easily adapted for any models with at most 10 lines of code change as shown in Appendix D. We anticipate the simple yet effective DCP will serve as a robust baseline and facilitate future advancements for distributed training with variable sequence lengths.

## 2 CHALLENGES IN TRAINING VARIABLE LONG SEQUENCES

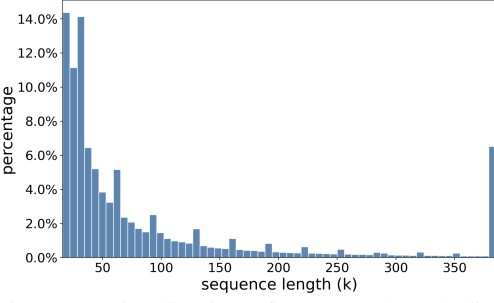

Figure 3: Visualization of sequence length distribution of Panda-80M.

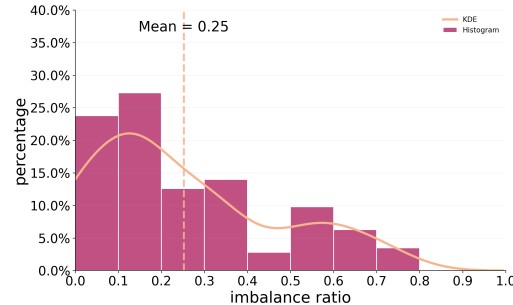

Figure 4: Imbalance ratio analysis across 140 datasets based on bucket parallel.

### 2.1 LARGE VARIANCE IN DATA LENGTH

The primary challenge is the large variance in sequence lengths in real-world datasets. For example, Figure 3 demonstrates it with the Panda-70M (Chen et al., 2024a) dataset, which contains 70 million videos. As a result, the training system must handle extreme diversity in workload.

A naive solution is to group data of similar lengths into one training step. However, this approach harms model quality because it disrupts the independent and identically distributed sampling crucial for stable convergence (Waltz, 1984; Brooks et al., 2024). Therefore, an effective training system must handle this extreme length diversity without compromising statistical efficiency.

## 2.2 Communication Cost for Sequence Parallel

For the longest sequences in a dataset, a large SP size is inevitable to avoid out-of-memory errors. This maximum required parallelism often dictates the global SP configuration for the entire training.

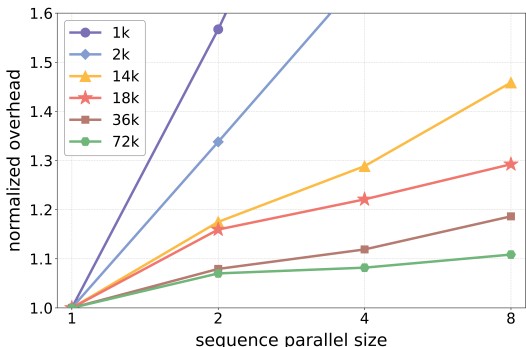

Figure 5: Weak scaling communication overhead for different sizes of sequence parallel on various sequence length for Transformer-1D.

**Large SP is necessary for long sequences.** For the longest sequences, a large SP size is inevitable to avoid out of memory error, and decides the global sequence parallel size.

**Large SP is harmful for shorter sequences.** This large SP becomes a major bottleneck for shorter sequences as shown in Figure 5, which often dominate datasets. For these sequences, the computation on each device is less, but the fixed cost of communication required to exchange information between devices remains high. This communication overhead can easily dominate the total processing time, drastically reducing training efficiency.

This creates a dilemma: a large, fixed SP size is wasteful for the common short sequences, while a small size cannot accommodate the necessary long sequences.

## 2.3 Workload Imbalance for Distributed Training

When using Data Parallelism (DP), the high variance in sequence lengths causes severe workload imbalance as shown in Figure 4. In a DP system, each worker processes a different batch of data, but all workers must synchronize before starting the next step.

Due to length variation, a worker assigned a batch of long sequences will take significantly longer to complete its computation. In contrast, workers with short sequences finish quickly and are forced to sit idle, waiting for the slowest worker to complete its task. This effect leads to severe under-utilization of computational resources, as expensive accelerators waste cycles waiting. The overall training throughput is bottlenecked by the slowest worker in each step, a problem that is exacerbated as the number of workers in the cluster grows.

## 2.4 Trade-off between Balance and Communication

An intuitive approach to mitigate workload imbalance is to adjust batch sizes statically: use smaller batches for long sequences and larger ones for short sequences to equalize the processing time per step. However, this strategy is ineffective and introduces a costly new trade-off.

**Small batch size for long sequences.** For the long sequences, memory constraints already force the batch size to the minimum, so it cannot be reduced further. Even if there is a few batches for long sequences, after they reduce the batch size, for all other sequences, they need to use a batch size smaller than what the GPU can hold leads to the under-utilization of expensive hardware.

**Communication overhead for reduced batch sizes.** More critically, this approach creates a new bottleneck. By using smaller batches to balance workload in each, the total number of training steps needed to process the dataset increases substantially. Since every iteration requires a costly global synchronization step to collect all parameters' gradients, this strategy trades per-iteration workload imbalance for a massive increase in total communication costs. The result is often no net performance gain, or even a regression.

> *Takeaway: **workload imbalance**, **communication cost**, and **computation efficiency** are the three key factors to effectively train variable long sequences.*

# 3 DATA-CENTRIC PARALLEL

## 3.1 OVERVIEW

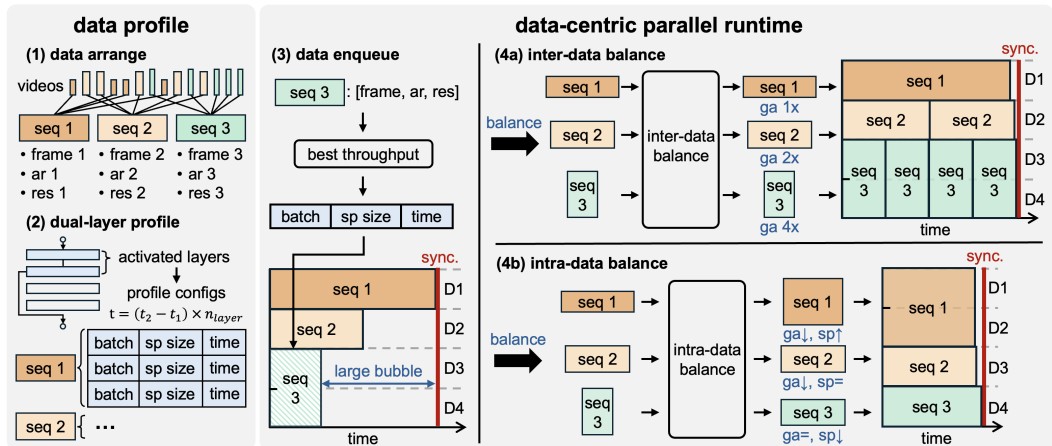

Figure 6: The overview of DCP. It first collects speed and memory cost by fast dual-layer profile. Then maximizes efficiency by dynamically adjusting parallel size, batch size, and gradient checkpointing based on sequence length.

As shown in Figure 6, Data-Centric Parallel (DCP) dynamically adjust the parallel, batch size and gradient checkpointing to maximize the efficiency. Specifically, we first group different sequences in to several groups with similar sizes. And for each group, we use a fast profiling to get the speed and memory cost for different batch sizes and sequence parallel sizes. Then, we apply one of the following two strategies to dynamically adjust the settings of based on data:

**DCP-inter** optimize load balance by gradient accumulation. Prior to each iteration, it selects the optimal batch size and sequence parallel size for the incoming batch's sequence length based on profile. Instead of reducing batch size, this method enable better balance with no cost.

**DCP-intra** further exploits dynamic recomputation for better speed. This is motivated by our analysis that for short sequences, gradient checkpointing introduces considerable unnecessary computational overhead. Based on this insight, DCP-intra implements a strategic trade-off: it partially deactivates gradient checkpointing. To manage the increase in memory consumption, it then dynamically adjusts sequence parallel and batch size with ignorable cost.

## 3.2 PROBLEM FORMULATION

Given $n$ batches $\{d_1, d_2, ..., d_n\}$ with variable sequence lengths $\{s_1, s_2, ..., s_n\}$ for each training iteration, DCP dynamically adapts training configuration of each batch according to its sequence length to improve the training throughput. The key is to jointly **improve the throughput** and **balance the execution time** for different sequence lengths in each training iteration. Formally, it is translated to two optimization targets:

$$\max\left(\frac{b_i * s_i}{p_i * T(d_i)}\right), \text{s.t.} \sum_{i=1}^{n} p_i = W, \tag{1}$$

$$\min \sum_{i=1}^{n} (\max_{j=1}^{n} T(d_j) - T(d_i))/W. \tag{2}$$

where $b_i$, $p_i$ and $T(n_i)$ are the batch size, sequence parallel size, and execution time for $n_i$, and $W$ is the total number of GPUs.

## 3.3 DCP-INTER

DCP-inter first determines the best parallel settings for each sequence length based on profiling. It then exploits gradient accumulation to balance the execution time across data batches in each itera-

tion instead of reducing batch size. Such simple but effective strategy maintains a high throughput for slow batches and fill the idle time of fast batches by running multiple batches.

Given $n$ batches, DCP-inter aims at finding the number of accumulation steps $g_i$ for each batch based $T(d_i)$ to meet balance and throughput. $T(d_i)$ is obtained by a fast profiling to be throughput-optimal for each $b_i$. To achieve optimal workload balance, Equation 2 can be rewritten as:

$$\min \sum_{i=1}^{n} \left( \max_{j=1}^{n} (g_j * T(d_j)) - g_i * T(d_i) \right)/W. \tag{3}$$

To search for $g_i$ meeting this target, DCP-inter iterates all possible number of accumulation steps for every batch and evaluates each combination of $g_i$ for $n$ batches using Equation 3 as the performance model to find the optimal setup for the $n$ batches in each iteration, detailed in Appendix C.

### 3.4   DCP-INTRA

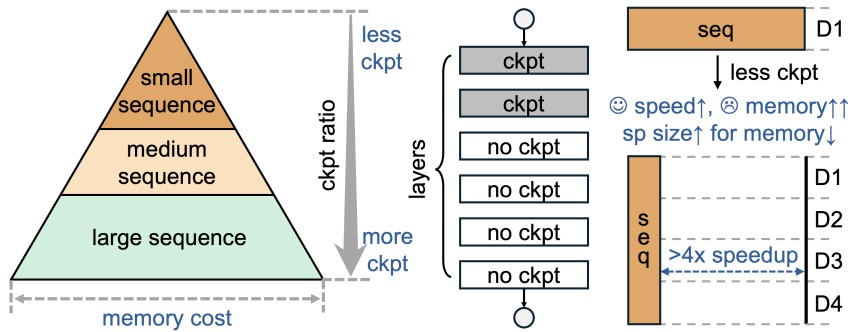

Figure 7: The overview of DCP-intra. It applies less recomputation for shorter sequences and increase sequence parallel size for extra memory cost.

For long sequence training, the activation checkpointing strategy is often fully applied to reduce the memory cost of GPU (Yuan et al., 2024). However, with variable sequence lengths, we have spare GPU memory to enable less recomputation layers for shorter sequences to reduce the execution time $T(b_i)$ Also, with larger sequence parallel size, the memory cost per GPU decreases, leaving more space for less recomputation to improve the throughput in Equation 1. Based on the two observations, we proposes DCP-intra.

For a $L$ layer model, the execution time $T(b_i)$ and the memory overhead $M(b_i)$ for $b_i$ are:

$$T(d_i) = T_f(d_i) + T_b(d_i) + T_f'(d_i) + T_C, \tag{4}$$
$$T_f'(d_i) = r_i * t_f(d_i), \tag{5}$$
$$M(d_i) = (L - r_i) * m(d_i) + L * m_{inp}(d_i) + M_C, \tag{6}$$

where $T_f(b_i)$ and $T_b(b_i)$ are the forward and backward execution time, and $T_C$ is the constant time cost by other components like embedding, loss computation, etc. $T_f'(b_i)$ is the recomputation execution time which denotes recomputing the forward passes of $r_i$ layers before the backward pass, with each layer using $t_f(b_i)$ time.

The memory overhead $M(b_i)$ comprises of three terms. The first is the memory overhead costed by layers that does not recompute. These layers need to stash all the intermediate activations with a cost of $m(b_i)$ for every layer for backward. The second is the necessary memory overhead for input activations $m_{inp}(b_i)$ for each layer. The third term $M_C$ includes a constant memory costed by model parameters, optimizers and intermediate buffers (Rajbhandari et al., 2020; Yuan et al., 2024).

The procedure is described in Algorithm 1. To reduce the execution time, for each sequence length $s_i$, it first runs a fast profiling for all possible batch size $b_i$ and sequence parallel size $p_i$ to get the execution time $T(d_i)$ and the memory overhead $M(d_i)$ with $r_i$ set to $L$. Then, according to Equation 6, it computes the minimal $r_i$ under the GPU memory capacity $M$ as the optimal recomputation for $b_i$ and $p_i$. Finally, we can compare the throughput according to Equation 1 for all possible batch size and parallel size to obtain the throughput-optimal settings.

---

**Algorithm 1** DCP-intra algorithm

---

    **Input:** Memory capacity $M$
    **Input:** Model layers $L$
    **Input:** Candidates $C$ from profiling $(b, p)$: $(t_f, m, T_{(b,p)}, M_{(b,p)})$
    **Output:** Optimal batch size $b_i$, sequence parallel size $p_i$, recompute layer $r_i$
1: $min\_time \leftarrow \infty$
2: Optimal result $(b_i, p_i, r_i) \leftarrow$ None
3: **for** $(b, p) \in C$ **do**
4:      $(t_f, m, T_{(b,p)}, M_{(b,p)}) \leftarrow C[(b, p)]$
5:      **for** $r \leftarrow 1$ to $L$ **do**      ▷ Calculate minimal recompute layer $r$ with spare GPU memory
6:          $M_{opt} \leftarrow M_{(b,p)} + r * m$
7:          **if** $M_{opt} \leq M$ **then**
8:              break
9:          **end if**
10:      **end for**
11:      $cur\_time \leftarrow T_{(b,p)} - r * t_f$      ▷ Calculate execution time $cur\_time$ with more memory
12:      **if** $cur\_time < min\_time$ **then**
13:          $min\_time \leftarrow cur\_time$
14:          $(b_i, p_i, r_i) \leftarrow (b, p, r)$
15:      **end if**
16: **end for**

---

### 3.5 DUAL LAYER PROFILING

DCP-intra and DPC-inter rely on a profiling process to set up an initial configurations. Based on the observation that a transformer model often consists of multiple repeating layers, the profiling process only run two layers for each batch size and sequence parallel size. The first layer is used to warm up, and the forward execution time $t_f(b_i)$, backward execution time $t_b(b_i)$ and memory overhead $m(b_i)$ for the second layer are recorded to estimate these metrics for the whole model. The result also includes the constant time $T_C$ and memory overhead $M_C$. With the four parts, the full execution time is estimated as $L * (2 * t_f(d_i) + t_b(d_i)) + T_C$. The total memory overhead is estimated by Equation 6, with $r_i$ set to $L$.

## 4 EXPERIMENTS

### 4.1 EXPERIMENT SETUP

**Models.** We test on two typical transformer architectures, Transformer-1D and Transformer-2D. Transformer-1D model is of 5B parameters, follow common architecture as in LLM (Dubey et al., 2024). Transformer-2D has 1.2B parameters, which applies attention sequentially across 2D sequences. This is common in models for extreme long sequences such as protein prediction (Jumper et al., 2021) and video generation (Zheng et al., 2024), detailed in Appendix B.

**Datasets.** To assess our method's performance across diverse sequence length distribution, we synthesize three datasets including: 1) short: dominated by short sequences; 2) balanced: balanced across a wide range of lengths; 3) long: dominated by long sequences, detailed in Appendix A.

**Baselines.** We takes bucket parallel (Esser et al., 2024) as our baseline, which fix parallel size and adjust batch size according to profiling results to balance workload.

**Testbed.** Our experiments are conducted on a cluster of NVIDIA H200 GPUs. Each node contains 8 GPUs connected via 900GB/s NVLink, with an 8×400Gbs InfiniBand network for internode communication. Although H200 GPUs have 141GB of memory, we limit usage to 80GB to demonstrate our method's generalizability to more common GPUs like the H100 or A100.

**Implementation details.** We adopt DeepSpeed Ulysses (Jacobs et al., 2024) for sequence parallel and ZeRO-1 (Rajbhandari et al., 2020) for data parallel, and enable Flash Attention (Dao et al., 2022) for all experiments. Note that our methods are compatible with all sequence parallel methods.

## 4.2 END-TO-END EVALUATION

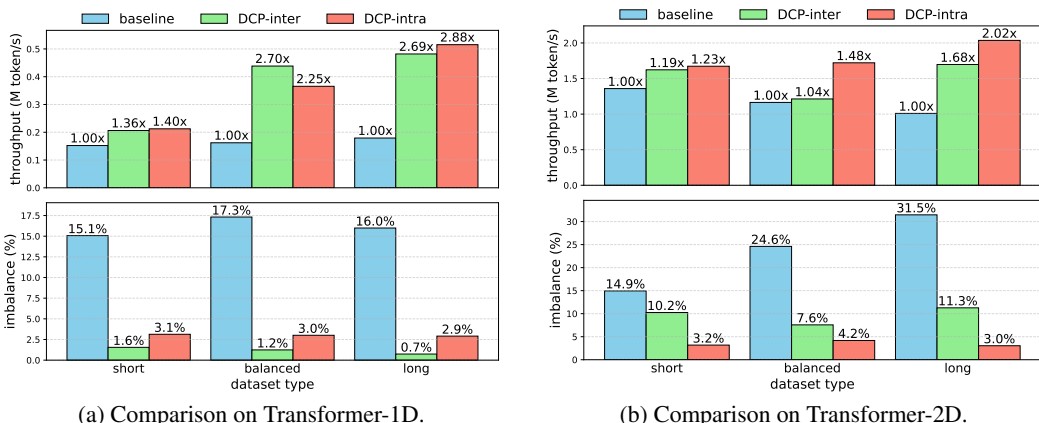

(a) Comparison on Transformer-1D.

(b) Comparison on Transformer-2D.

Figure 8: End-to-end throughput and imbalance comparison between different approaches for training Transformer-1D and Transformer-2D with 32 GPUs across 3 datasets.

**Throughput.** presents an end-to-end comparison of DCP-inter and DCP-intra methods against the baselines. The evaluation is conducted on Transformer-1D and Transformer-2D models using 32 H200 GPUs across three distinct synthesized datasets.

Based on the results, we make the following observations: 1) DCP-inter consistently outperforms the baseline, delivering speedups of up to $2.70\times$ and $1.68\times$ for the two models, respectively. 2) DCP-intra, which leverages dynamic checkpointing, provides additional performance gains over DCP-inter, achieving a total speedup of up to $2.88\times$. 3) The improvements of our methods are most pronounced on datasets dominated by shorter sequences.

**Imbalance.** As shown in Figure 8, our methods significantly reduce the workload imbalance observed across training tasks. We define the imbalance ratio as the average proportion of time a GPU is idle while waiting for computations on other devices. The results indicate that the baseline method suffers from a high degree of imbalance that also change significantly with the data distribution. In contrast, our methods maintain a consistently low imbalance ratio across all conditions, which is a direct result of our flexible strategy.

**Effect of datasets distribution.** Our methods achieve more performance gains on datasets dominated by shorter sequences. Because our method mitigates these issues by reducing sequence parallel, which lowers communication costs. Furthermore, unlike bucket parallel reduces the batch size for balance, our method maintains a full batch size, leading to better computation efficiency.

**Effect of model architecture.** Our method performs better on Transformer-1D because the computation time is significantly longer as it uses 1D attention compared with 2D attention, so the imbalance will be more severe.

## 4.3 SCALING ABILITY

Scaling ability is of critical importance to training variable long sequences, as the communication cost, and especially workload imbalance will increase significantly as the scale of devices increases. Because larger-scale systems often encounter a greater diversity of sequence lengths.

As shown in Figure 9, we present a weak-scaling comparison of our method against the baselines. The results yield two primary observations: 1) The bucket parallel baseline exhibits a significant decay in throughput as it scales, due to its inherent trade-offs between load balance and computational efficiency. 2) In contrast, DCP demonstrates substantial scalability improvements, achieving near-linear scaling on both Transformer-1D and Transformer-2D models.

The consistent scaling performance of DCP establishes it as a highly effective and robust solution for training variable long sequences at scale, regardless of model architecture and data distribution.

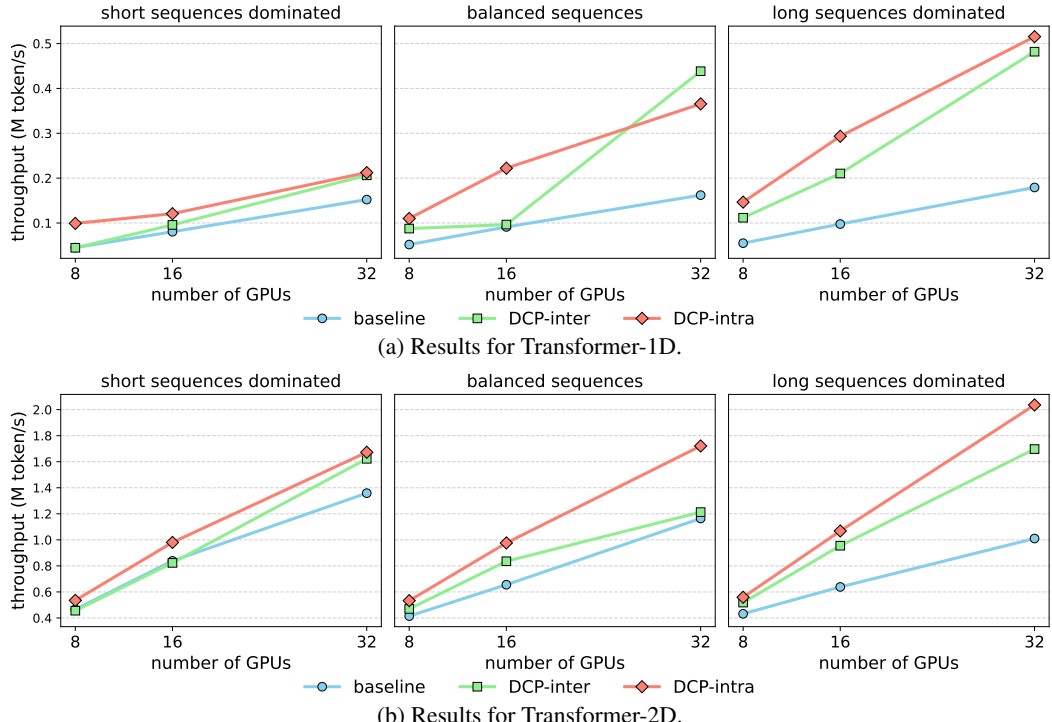

(a) Results for Transformer-1D.

(b) Results for Transformer-2D.

Figure 9: Scalability evaluation of different methods with different models.

## 4.4 ABLATION STUDY

As shown in Figure 10, we evaluated the speedup of DCP-intra compared with DCP-inter across various sequence lengths. The results demonstrate that for most sequence lengths, our method achieves 20-25% speedup for sequence length less than 200k. This performance gain directly corresponds to the overhead of gradient checkpointing, which our method effectively eliminates. Furthermore, by allowing reasonable adjustments to the batch size and sequence parallel degree, our method can address extra memory cost with ignorable computational overhead.

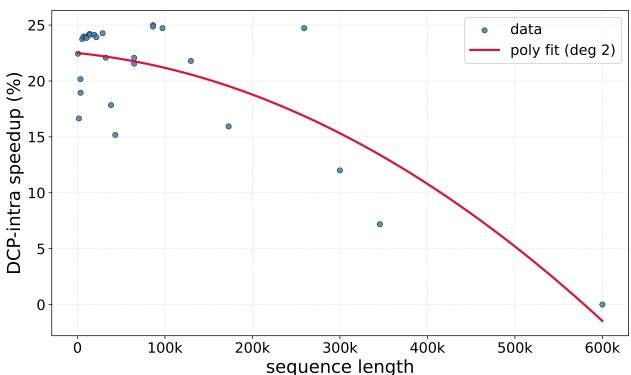

Figure 10: The speedup of DCP-intra over DCP-inter across various sequence lengths on Transformer-1D.

## 5 RELATED WORK

### 5.1 PARALLEL TRAINING

**Data parallelism** is a commonly adopted technique to enable distributed training of neural networks (Xing et al., 2015). It replicates the model to different workers and split the global data batch to multiple batches. With the advent of transformer models (Brown et al., 2020; Touvron et al., 2023; Yang et al., 2024), the memory capacity of a single GPU cannot accommodate the high memory demand. ZeRO (Rajbhandari et al., 2020) reduces the memory redundancy from optimizer states, gradients and model parameters across different workers to enable large scale model training.

**Sequence parallelism** is used for distributed training with long sequence length. DeepSpeed Ulysess (Jacobs et al., 2024) partitions the input activations, query, key and value for the attention

module along the sequence dimension, with each sequence parallel worker possessing one split. After getting the local output, another all-to-all operation is used to recover the attention heads and split along sequence dimension. Ring attention also partitions the query, key and value along the sequence dimension (Li et al., 2023; Fang & Zhao, 2024). To get the attention output, the key and value splits of different workers are exchanged via p2p communication in a ring pattern to perform and accumulate partial computation with the local query split. Sequence parallelism of Megatron-LM (Korthikanti et al., 2023) splits both activations and model parameters across sequence parallel workers.. It uses all-gather operation to recover the full input sequence and reduce-scatter operation to reduce and split the output sequence.

## 5.2 ACTIVATION CHECKPOINTING

Activation checkpointing trades off memory cost against computation during model training (Chen et al., 2016). Concretely, during the forward pass for a sequence of modules of a model, certain intermediate activations need to be stashed for gradient computation during backward pass. Activation checkpointing only stashes the input activation for the sequence of modules, and computes the forward pass to materialize the activations needed for the backward pass. Existing works either utilize heuristics to design static strategies (Chen et al., 2016; Narayanan et al., 2021; Korthikanti et al., 2023), or automatically searches for adaptive strategies that are optimal in terms of certain optimization targets (Jain et al., 2020; Sun et al., 2024; Yuan et al., 2024).

## 5.3 DYNAMIC PARALLELISMS FOR VARIABLE SEQUENCE LENGTH

Variable sequence length emerges as a new topic for large language models. HotSPa fixes the total number of GPUs for a training task and switches the degree of data, sequence and pipeline parallelisms according to the sequence length at the runtime (Ge et al., 2024). Tenplex is designed for elasticity problem where the number of GPU may change at runtime to support dynamic update of parallelisms for long-running training jobs (Wagenländer et al., 2024). Both methods mainly focus on reducing the communication overhead for moving model parameters and optimizer states caused by the parallelism change. LoongServe targets on the inference task and enables elastic sequence parallelism to efficiently support variable sequence length (Wu et al., 2024). Similar to the two methods for training tasks, the major consideration for LoongServe is to relieve the communication overhead costed by moving KV cache during the change of sequence parallelism.

Despite their curated design for large language model training or inference, the variable sequence length issue on transformer model training is underexplored.

## 6 CONCLUSION

This paper introduces Data-Centric Parallel (DCP), a framework that resolves the trade-off between efficiency and ease-of-use when training deep learning models on variable long sequences. Unlike traditional methods that force a choice between inefficient static configurations and complex, model-specific code, DCP breaks this by letting the data itself drive the runtime. By dynamically adjusting settings like parallel size and gradient accumulation based on each batch's sequence length, DCP ensures more effective hardware utilization. Our empirical results demonstrate up to a 2.88x speedup on 32 H200 GPUs. A key advantage is its simplicity; DCP can be integrated into any model with only ten lines of code, promoting wide adoption. We believe this effective method will serve as a robust baseline for distributed training and facilitate future advancements in this area.

**Limitation.** This method is currently subject to two primary limitations. First, its application is restricted to Transformer-based models. Extending DCP to other architectures would require new schedulers and profiling strategies. Second, it's designed for a single model architecture and cannot be applied to systems of multiple, distinct networks.

**Future works.** Future work could enhance this method by developing predictive models to proactively select a wider range of dynamic runtime parameters, moving beyond the current profiling based adjustments. Furthermore, exploring finer-grained, intra-batch parallelism could offer more efficiency with highly diverse sequence lengths, further minimizing padding-related overhead. The scope of DCP can also be broadened to other architectures.

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

# Training Variable Long Sequences with Data-Centric Parallel

# Appendix

We organize our appendix as follows:

- Section A: Dataset configuration.
- Section B: Model settings.
- Section C: Implementation details of DCP-inter.
- Section D: API usage.
- Section E: Ethics statement.
- Section F: Reproducibility statement.
- Section G: LLM usage.

## A    DATASET CONFIGURATION

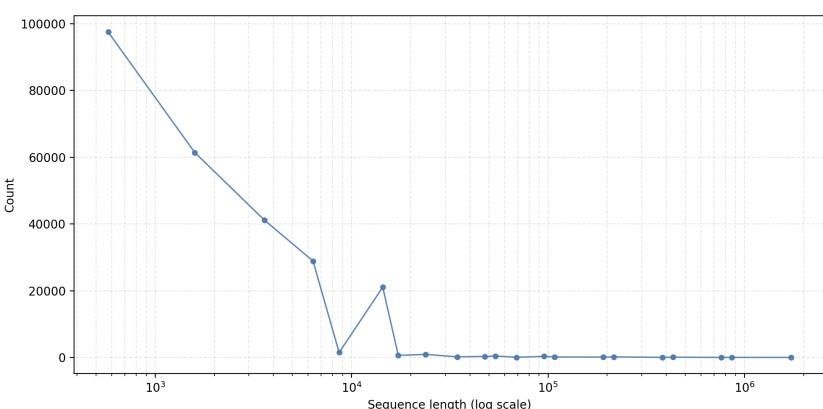

Figure 11: Distribution of short sequences dominated dataset.

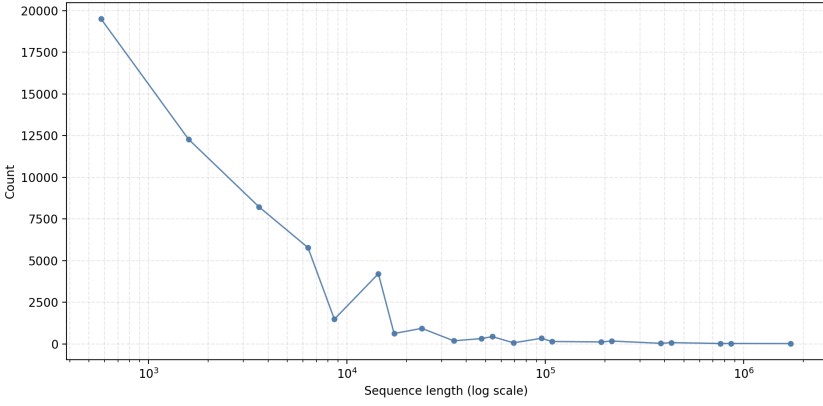

Figure 12: Distribution of balanced dataset.

The sequence length distributions for our three evaluation datasets—short-sequence-dominated, balanced, and long-sequence-dominated—are visualized in Figures 11, 12, and 13, respectively.

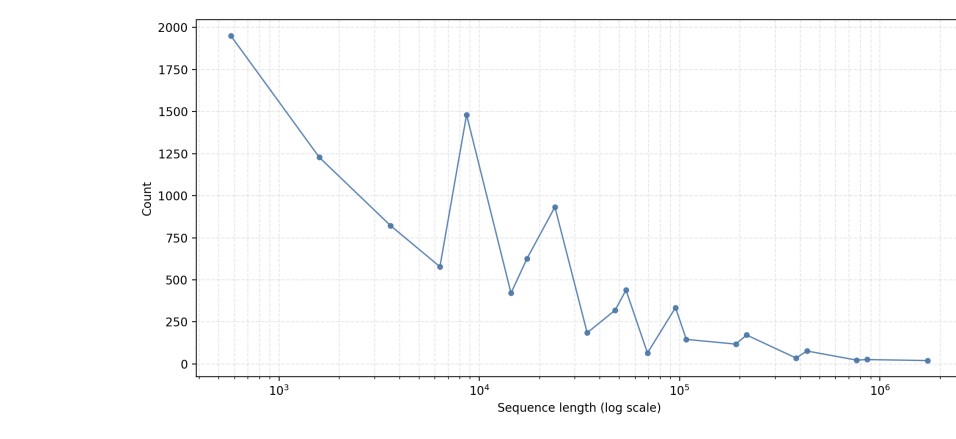

Figure 13: Distribution of long sequences dominated dataset.

We define dataset dominance by computational cost, not by the raw count of sequences. Consequently, in the long-sequence-dominated dataset, the running time contributed by a few long sequences is orders of magnitude greater than that of the more numerous short sequences, thereby dictating the overall training time.

## B   MODEL SETTINGS

Table 1: Model settings of Transformer-1D and Transformer-2D.

| Model Name | Parameter Number | Layers | Hidden States | Attention Heads |
|---|---|---|---|---|
| Transformer-2D | 1.2B | 28 | 1152 | 16 |
| Transformer-1D | 5B | 42 | 3072 | 48 |

## C   IMPLEMENTATION DETAILS OF DCP-INTER

There are two additional considerations for DCP-inter when setting the number of gradient accumulation steps.

First, as this search procedure should run for each training iteration at runtime, it should be lightweight and fast to ensure not slow down the actual training process.

Second, while extremely large number of accumulation steps can make the workload imbalance negligible, this will also leads to unstable convergence speed and longer training time (You et al., 2020; Luo et al., 2023). Therefore, DCP-inter limits $g_i$ to a range $[G_{min}, G_{max}]$ to relieve the two problems, detailed described

$G_{max}$: Within an iteration, find the batch with the longest processing time. The maximum time for this iteration is calculated by multiplying the duration of that batch by this value.

$G_{min}$: If the gas for every batch within an iteration is less than this value, then the gas for each batch is scaled up by a factor of $[G_{min}$ / gas].

## D   API USAGE

## E   ETHICS STATEMENT

This work adheres to the ICLR Code of Ethics. In this study, no human subjects or animal experimentation was involved. All datasets used, including Panda-70M, were sourced in

```python
from dcp.profiler import set_manager

# 1. set the dcp manager
dcp_manager = set_manager(model)

# 2. modify the training loop
for epoch in range(epochs):
    # insert 2 lines to get the data iterator
    if dcp_manager.need_profile():
        dcp_manager.init_profiler()
    dataloader_iter = dcp_manager.get_data_iter()

    for step, batch in enumerate(dataloader_iter):
        # Insert 2 lines to change sequence
        # parallel size and enable profile
        dcp_manager.optimize_dynamics(batch, model)
        with dcp_manager.profile(batch, model, gas):
            batch = batch.cuda()
            loss = model(batch)
            model.backward(loss)
            model.step()

# 3. recompute the model
for layer in model.layers:
    x = dcp_manager.recompute(layer, x)

# 4. modify communication ops
torch.distributed.all_to_all(x, group=dcp_manager.group)
```

Figure 14: The API usage to enable our methods within 10 lines of code.

compliance with relevant usage guidelines, ensuring no violation of privacy. We have taken care to avoid any biases or discriminatory outcomes in our research process. No personally identifiable information was used, and no experiments were conducted that could raise privacy or security concerns. We are committed to maintaining transparency and integrity throughout the research process.

## F    REPRODUCIBILITY STATEMENT

We have made every effort to ensure that the results presented in this paper are reproducible. All code and datasets have been made publicly available in an anonymous repository to facilitate replication and verification. The experimental setup, including training steps, model configurations, and hardware details, is described in detail in the paper.

Additionally, our code will be publicly available, ensuring consistent and reproducible evaluation results.

We believe these measures will enable other researchers to reproduce our work and further advance the field.

## G    LLM USAGE

Large Language Models (LLMs) were used to aid in the writing and polishing of the manuscript. Specifically, we used an LLM to assist in refining the language, improving readability, and ensuring

clarity in various sections of the paper. The model helped with tasks such as sentence rephrasing, grammar checking, and enhancing the overall flow of the text.

It is important to note that the LLM was not involved in the ideation, research methodology, or experimental design. All research concepts, ideas, and analyses were developed and conducted by the authors. The contributions of the LLM were solely focused on improving the linguistic quality of the paper, with no involvement in the scientific content or data analysis.

The authors take full responsibility for the content of the manuscript, including any text generated or polished by the LLM. We have ensured that the LLM-generated text adheres to ethical guidelines and does not contribute to plagiarism or scientific misconduct.

