# OpenReview forum: "Training Variable Long Sequences With Data-centric Parallel"
_ICLR.cc/2026/Conference — Submitted to ICLR 2026_

### Official Review · Reviewer_Dwa4 · 2025-10-31

**Soundness:** 2
**Presentation:** 2
**Contribution:** 2
**Rating:** 2
**Confidence:** 5

**Summary:**

This work introduces Data-Centric Parallel (DCP), which addresses the workload imbalance caused by the variable sequence lengths in large language model (LLM) training. There are two variants of DCP. Based on the sequence length information, DCP-inter dynamically adjusts the runtime sequence parallel (SP) degree and the number of accumulation steps for each micro-batch to balance the execution time, and DCP-intra further adjusts the recomputation cost for each sequence. Experimental results show that DCP achieves up to 2.88 times of speedup compared to the fixed parallel baseline.

**Strengths:**

(S1) The workload imbalance caused by the variable sequence lengths is an essential challenge in LLM training. It would be great to tackle such a challenge.

(S2) Two data-centric parallel approaches are proposed, both supporting dynamic adjustment to the parallel training configuration with data-awareness.

(S3) The speedup compared to the baseline is substantial.

**Weaknesses:**

(W1) My major concern is the limited novelty and technical contribution. Existing works have analyzed the workload imbalance/heterogeneity due to the variable sequence lengths, and proposed dynamic sequence/context parallel approaches for variable-length training [1,2,3]. FlexSP [1] and WLB-LLM [2] formulate optimization problems to determine the parallel degrees and data batching/packing (which affects the number of gradient accumulation steps). ByteScale reduces the memory required by applying selective offloading to long sequences, which is similar to DCP-intra’s approach of applying activation checkpointing. Currently, the paper does not discuss the differences to existing works, nor conduct any empirical comparison.

[1] FlexSP: Accelerating Large Language Model Training via Flexible Sequence Parallelism. ASPLOS 2025. https://arxiv.org/pdf/2412.01523, https://dl.acm.org/doi/10.1145/3676641.3715998 \
[2] WLB-LLM: Workload-Balanced 4D Parallelism for Large Language Model Training. OSDI 2025. https://arxiv.org/pdf/2503.17924, https://www.usenix.org/conference/osdi25/presentation/wang-zheng \
[3] ByteScale: Communication-Efficient Scaling of LLM Training with a 2048K Context Length on 16384 GPUs. SIGCOMM 2025. https://arxiv.org/pdf/2502.21231, https://dl.acm.org/doi/10.1145/3718958.3754352

(W2) It is unclear why there are two optimization targets: Eq (1) aims at maximizing throughput and is used in DCP-inter, while Eq (2) aims at minimizing execution time and is used in DCP-intra. For training, throughput is often defined as the number of tokens/sequences trained per second, and therefore it is computed as the size of a global training batch divided by per-iteration execution time, implying that these two optimization targets are equivalent. In addition, according to Section 3, the profiling is done for each batch. However, during training, the sequence length distribution should vary across batches, so it is unclear how the profiled results can adapt to the unknown distribution.

(W3) The experiments only take bucket parallel as the baseline, which is a weak baseline. Packed parallel (depicted in Figure 1 of the paper) and the existing works mentioned above should also be considered as baselines.

(W4) The experimented models are quite small (1.2B and 5B) given the cluster setup of 32 H200 GPUs. IMHO, a possible reason is that SP requires each worker holding the complete model, which restrict the model size that DCP could accommodate. Following this, it is also expected to discuss how DCP can be incorporated with model parallel approaches to accommodate larger models.

**Questions:**

(Q1) Please compare DCP with existing works.

(Q2) Please clarify the problem formulation and cost profiling.

(Q3) Please take more suitable baselines into account.

(Q4) Please consider experimenting on larger models.

---

### Official Review · Reviewer_F3HF · 2025-11-01

**Soundness:** 2
**Presentation:** 3
**Contribution:** 3
**Rating:** 4
**Confidence:** 5

**Summary:**

This paper presents Data-Centric Parallel (DCP), a new framework designed to improve the efficiency of distributed training for models that process variable-length sequences, such as Transformers. The authors identify a core bottleneck in training workloads: significant imbalance in computation time and memory usage across batches due to sequence length variation. DCP addresses this by dynamically adjusting runtime configurations—parallelism degree, batch size, gradient accumulation, and recomputation—based on each batch's sequence length. The paper introduces two complementary strategies: DCP-inter, which uses adaptive gradient accumulation to balance workloads across devices, and DCP-intra, which further reduces computation time by adjusting gradient checkpointing for short sequences.

**Strengths:**

1. The problem is well-motivated and practically relevant, addressing an important gap in distributed training efficiency for variable-length sequence models.
2. The proposed data-centric parallel framework is conceptually simple yet novel -- it shifts control from static runtime configurations to data-driven adaptation, which is elegant in its generality.

**Weaknesses:**

1. The main experiments rely solely on synthetic datasets, which undermines the claimed generalizability. Although synthetic data provides controlled variation in sequence length, it does not capture real-world statistical properties, data irregularities, or convergence dynamics that could reveal DCP's true robustness.
2. The evaluation uses only two relatively simple Transformer models (1D and 2D). Other Transformer variants are not included.
3. The paper lacks end-to-end training convergence analysis on realistic datasets.

**Questions:**

1. Fig. 5: How exactly is the communication cost measured? Is it based on profiling or estimations?
2. Since DCP dynamically changes training configurations, how does this grouping or dynamic adjustment affect the data distribution seen by the optimizer? Have you measured convergence behavior to ensure it remains consistent with static baselines? Or do you have a theoretical guarantee that this won't change the IID assumption in training?
3. The paper introduces Panda-70M in the motivation section but does not use it for actual evaluation. Why does the evaluation section use synthetic datasets instead?
4. Do users need to manually decide whether to use DCP-inter or DCP-intra, or can the system automatically choose based on profiling data? Also, could these two strategies be integrated into one?
5. How was DCP prototyped? Was it implemented on top of Megatron-LM or DeepSpeed? Is the bucket parallel baseline built on the same framework to ensure a fair comparison?

Nits: Please carefully proofread your paper.
1. Page 2 Figure 3 caption: should be Panda-70M?
2. Page 2 Line 65: an additional "and"
3. Page 2 Line 66: preventing->prevents
4. Page 5 Line 47: Lack a period after T(b_i)

---

### Official Review · Reviewer_NNhB · 2025-11-01

**Soundness:** 3
**Presentation:** 3
**Contribution:** 3
**Rating:** 6
**Confidence:** 1

**Summary:**

This paper presents Data-Centric Parallel (DCP), a runtime framework for efficiently training models on variable-length sequences.
Standard sequence-parallel training uses a fixed configuration, which leads to imbalance across GPUs and wasted computation when sequence lengths vary widely.
DCP dynamically adjusts parameters such as the degree of parallelism, batch size, gradient accumulation, and checkpointing to balance workloads. Experiments on Transformer models (1D and 2D) show up to 2.9× throughput improvements and reduced idle GPU time.

**Strengths:**

1. The paper addresses a practical and under-explored bottleneck in training models with variable sequence lengths, a real issue for LLMs and multimodal systems. The paper is clearly written and well structured.
2. The proposed DCP framework is lightweight, easy to adopt, and integrates smoothly with existing training pipelines.
3. The presentation and empirical validation are consistently strong across different GPU scales and model sizes.

**Weaknesses:**

1. Evaluation scope is limited to synthetic datasets; no results on realistic workloads such as video or text corpora.
2. Limited analysis on training convergence or accuracy: The paper mainly reports throughput and utilization improvements, but provides no evidence that the method preserves training convergence or final task accuracy.

**Questions:**

1. Does dynamically changing gradient accumulation or checkpointing ever affect convergence or final model quality?

---

### Official Review · Reviewer_cuBK · 2025-11-03

**Soundness:** 3
**Presentation:** 4
**Contribution:** 3
**Rating:** 4
**Confidence:** 3

**Summary:**

This paper proposes a small modification to training that increases throughput in settings with highly variable context lengths. The main ideas is to adaptively adjust the parallelism parameters as a function of the context length, saving overhead when extreme sharding is not needed.

**Strengths:**

- The method is extremely simple yet seemingly effective, providing a strong baseline for training efficiency improvements
- The presentation is very clear and intuitive, and the paper was enjoyable to read
- The method shows promising performance on synthetic datasets.

**Weaknesses:**

- As far as I can tell, all the experiments are done on synthetic data - it would be helpful/more powerful to show benefits on real-world datasets. For example, the paper uses the Panda-70M dataset as motivation, but I couldn't find any experimental results there.
- I am not an expert in this field, but I am sure this is not the first paper to propose a way of dealing with variable-length sequences (even as a non-expert, I have heard of HotSPa) - some comparisons to these methods (or an explanation for why a comparison wouldn't make sense) would be helpful. Even a comparison to packed parallel or other baselines would strengthen the results.
- Appendix C seems incomplete, or at least does not contain what I expected following a forward reference from L227.
- Figure 9 non-monotonicity suggests there's a decent amount of inter-run variation - as a result it would be good to have error bars/uncertainty quantification around the speedups. Seems like "2.88x" is probably an instance of false precision.
- [Minor] The poly-fit in Figure 10 is not super convincing.

**Questions:**

- How is this different from HotSPa as mentioned in the related work (Ge et al.)?
- How does the relative performance of these methods change as a function of GPU interconnect, and GPU memory?

---

### Meta-Review · Area_Chair_5DWe · 2026-01-11

**Summary:**

Data-Centric Parallel tunes execution in terms of parallelization, batch size, gradient accumulation, and checkpointing/recomputation/rematerialization depending on the input data and specifically its sequence length. Strengths include addressing a real and current problem in distributed training for sequence models (and language models in particular), simplicity of implementation and adoption, clear exposition, and speed-up on synthetic datasets. Weaknesses include only synthetic data without real workloads of text of video sequence data, insufficient novelty and context w.r.t. existing work on efficient parallelization and variable length computation, and limited attention to the effect on performance (i.e. loss or accuracy, not computational performance).

Four experts are divided between rejection (cuBK: 4, F3HF: 4, Dwa4: 2) and acceptance (NNhB: 6). The vote for acceptance has extremely low confidence by its self-reporting and in discussion. In terms of experiments, the lack of real data and the missing care to check for potential impact on optimization indicate rejection, and in terms of scholarship, the lack of comparisons and references to existing work on variable sequence length likewise indicate rejection. No rebuttal is provided to address these issues among the others raised by reviewers. The meta-reviewer sides with rejection.

**Reviewer Concerns:**

- Missing comparison to existing work on efficient computation of variable-length sequences (Dwa4, cuBK): unresolved.
- Only experiments with synthetic data (cuBK, NNhB, F3HF): unresolved.
- Potential adverse impact on optimization and convergence (NNhB, F3HF): unresolved.
- Other smaller but nevertheless concrete issues with the implementation and comparison to existing methods as raised by the set of reviewers.

**Reviewer Scores:**

No change is expected because there is no rebuttal and there is agreement on multiple points by the reviewers.

---

### Decision · Program_Chairs · 2026-01-26

Reject